# Heat Shock Protein 90 (Hsp90) and Hsp70 as Potential Therapeutic Targets in Autoimmune Skin Diseases

**DOI:** 10.3390/biom12081153

**Published:** 2022-08-20

**Authors:** Stefan Tukaj, Krzysztof Sitko

**Affiliations:** Department of Molecular Biology, Faculty of Biology, University of Gdansk, 80-308 Gdansk, Poland

**Keywords:** heat shock proteins, Hsp90, Hsp70, skin diseases, autoimmunity, therapy

## Abstract

Over a hundred different autoimmune diseases have been described to date, which can affect every organ in the body, including the largest one, the skin. In fact, up to one-fifth of the world’s population suffers from chronic, noninfectious inflammatory skin diseases, the development of which is significantly influenced by an autoimmune response. One of the hallmarks of autoimmune diseases is the loss of immune tolerance, which leads to the formation of autoreactive lymphocytes or autoantibodies and, consequently, to chronic inflammation and tissue damage. The treatment of autoimmune skin diseases mainly focuses on immunosuppression (using, e.g., corticosteroids) but almost never leads to the development of permanent mechanisms of immune tolerance. In addition, current therapies and their long-term administration may cause serious adverse effects. Hence, safer and more effective therapies that bring sustained balance between pro- and anti-inflammatory responses are still desired. Both intra- and extracellular heat shock proteins (Hsps), specifically well-characterized inducible Hsp90 and Hsp70 chaperones, have been highlighted as therapeutic targets for autoimmune diseases. This review presents preclinical data on the involvement of Hsp90 and Hsp70 in modulating the immune response, specifically in the context of the treatment of selected autoimmune skin diseases with emphasis on autoimmune bullous skin diseases and psoriasis.

## 1. Introduction

Present in both prokaryotic and eukaryotic cells, highly evolutionarily conserved heat shock proteins (Hsps) are grouped into six major families: Hsp100, Hsp90, Hsp70, Hsp60 (chaperonins), Hsp40, and the small Hsps, according to their approximate molecular weight expressed in kilodaltons (kDa). Their classical ATP-dependent (foldase) or ATP-independent (holdase) chaperone activity is to assist in proper protein folding during translation, re-folding of denatured proteins, native protein stabilization, polypeptide transport, or degradation of misfolded proteins. Hsps can be synthesized constitutively or induced in the cell under stress conditions and are present in various cellular compartments. Multiple stress factors, e.g., hyperthermia, oxidative stress, ethanol, or UV radiation, can increase the production of well-characterized inducible Hsp90 and Hsp70 chaperones, which, in turn, interact with protein substrates (clients) and (co-)chaperones to participate in almost every cellular process, including inflammation [1,2,3,4]. In addition to their classical (canonical) role in the cell, some of these molecular chaperones (e.g., Hsp70) can be either passively (due to necrosis) or actively (via lipid vesicles) secreted into the extracellular space to mediate cell–cell communication via binding to several cell-surface receptors, including TLR4, CD91, LRP-1, SCARF1, and LOX-1 [5,6]. Moreover, some Hsps may form complexes with intracellular antigens, which are subsequently directed toward either the MHC class I or MHC class II associated pathways, leading to the activation of T lymphocytes [7]. As a result of findings described above, both intra- and extracellular Hsps have become the subject of interest of scientists in the context of the inflammatory and autoimmune process [8,9,10].

## 2. Heat Shock Protein 90 (Hsp90)

In humans, the Hsp90 family consists of four members (i.e., Hsp90α, Hsp90β, Grp94/gp96, and TRAP-1) localized in different cellular compartments. Structurally, all members of the Hsp90 family comprise a common domain structure consisting of the nucleotide-binding domain (NTD, N-terminal domain), a middle domain (MD) that has high affinity for co-chaperones and client proteins, and the carboxy-terminal domain (CTD) responsible for dimerization and oligomerization [11,12,13]. Hsp90α, so far the best described member of the Hsp90 family, is a stress-inducible mammalian cytosolic isoform. It is one of the key chaperones responsible for the biological activity of hundreds of its protein substrates, among which there are key signaling molecules (e.g., mitogen-activated protein kinase, MAP kinases) and transcription factors (e.g., nuclear factor-kappa B, NF-κB) that regulate important cellular processes, such as growth, cell cycle, and differentiation. Moreover, Hsp90 is involved in the stabilization of oncogenic proteins (e.g., Bcr-Abl, HER-2, EGFR, and HIF-1α); hence, the inhibitors of its chaperone activity are currently being tested as cancer treatment in advanced clinical trials [14,15]. In addition, it was shown that extracellular Hsp90α mediates tumor metastasis but can also contribute to physiological processes such as wound healing [16,17].

### Inhibition of Hsp90 Activity

The structure and chaperone activities of Hsp90 have already been thoroughly discussed in other review articles [18,19,20,21,22]. Here, we present a brief overview of this issue to provide the reader with a basic background for understanding the mechanism of action of Hsp90 inhibitors and their classification, which may be considered for use in the treatment of autoimmune skin diseases. The majority of discovered Hsp90 inhibitors that target the ATP-binding pocket in the N-terminal domain of Hsp90 have entered clinical trials (e.g., 17-DMAG, 17-AAG, and STA-9090); however, they have not been approved by the FDA due to high toxicity, poor pharmacokinetic profiles, or simply a lack of clinical efficacy [20]. The lack of effectiveness of this type of Hsp90 inhibitor is due to, inter alia, triggering a survival mechanism in cancer cells, referred to as heat shock response (HSR), driving heat shock factor 1 (HSF-1) activation [23]. In fact, activation of HSF-1 is a common feature of numerous cancer types, and its expression is associated with malignancy and mortality. Moreover, activation of HSF1-dependent chaperones (e.g., Hsp70, Hsp40, and Hsp27) participates in cancer cell growth and survival [10]. The above-mentioned insufficient therapeutic strategy in cancer diseases can be improved by using combined therapy. For instance, treatment with 17-AAG along with small interfering RNA (siRNA), knocking down Hsp90α, exhibited significant anticancer activity in glioma [24]. Additionally, Hsp90 siRNA was able to inhibit the proliferation, migration, and invasion of angiosarcoma cells [25]. Nevertheless, due to the significant role of Hsp90 in neoplastic diseases, the search for effective and safer inhibitors of this chaperone is still ongoing. For instance, isoform-selective inhibitors (e.g., Gamitrinib or Radamide) as well as regulators targeting the MD (e.g., Sansalvamide A, Kongesin A) or C-terminal region of Hsp90 (e.g., Novobiocin, Deguelin, Epigallocatechin-3-gallate (EGCG)) were reported as alternative strategies for developing potent and safer Hsp90 inhibitors for clinical use [18,19,20,21,22,26,27,28]. The latter group of Hsp90 inhibitors is an especially attractive chemotherapeutic approach, as they do not trigger an HSR [23] (Figure 1).

Importantly, numerous studies have shown that intracellular Hsp90 is involved in the activation of the innate and adaptive components of the immune response, thereby promoting an inflammatory/autoimmune response. An indirect contribution of Hsp90 in the autoimmune process was confirmed in experimental preclinical studies using Hsp90 inhibitors, belonging solely to the N-terminal binding type, which seem to be more attractive for the treatment of autoimmune diseases due to the activation of an HRS [10,23]. This anti-Hsp90 therapy has been successfully applied in murine models of encephalomyelitis, collagen-induced arthritis, adjuvant-induced arthritis, systemic lupus erythematosus (SLE), lipopolysaccharide-induced uveitis (EIU), and DSS-induced ulcerative colitis [10,29]. Mechanistically, Hsp90 inhibitors such as geldanamycin (GA) and its semi-synthetic derivatives (e.g., 17-DMAG or 17-AAG) bind to the N-terminal nucleotide binding pocket of Hsp90 with higher affinity than ATP, which drives Hsp90-dependent ‘clients’ to proteasomal degradation. It is believed that immunosuppressive activity of this type of Hsp90 inhibition may result from (i) the activation of HSF-1, which regulates the expression of multiple genes, including immunoregulatory Hsp70 and IL-10; (ii) the expansion of immunosuppressive lymphocytes (both T and B regulatory cells); (iii) the inactivation of NF-κB-dependent inflammatory/regulating factors, including TNF-α, IL-6, IL-8, and IL-17; or (iv) blockade of the cell signaling molecules, such as MAP kinase [10,14,29,30,31]. There is, however, an important issue to be resolved. Namely, will Hsp90 inhibitors, which already found use in the treatment of cancer and do not activate an HSR, also be effective in the treatment of autoimmune diseases? The basic mechanism and cellular consequences concerning the inhibition of Hsp90 in regard to autoimmune diseases are schematically presented in Figure 2.

## 3. Heat Shock Protein 70 (Hsp70)

The Hsp70 family of molecular chaperones (comprises 13 gene products in humans) represents one of the most ubiquitous classes of constitutively (e.g., Hsc70) or stress-induced (collectively termed Hsp70 or HSPA1) proteins [32]. Structurally, Hsp70 isoforms consist of three functional domains, including the nucleotide-binding domain (NBD), substrate-binding domain (SBD), and a C-terminal peptide-binding domain [33]. The Hsp70 is involved in a large variety of cellular processes, e.g., protein folding and remodeling, together with other molecular (co-)chaperones, including Hsp40 or Hsp90 [34]. While the inhibitors of Hsp70 chaperone activity are currently under investigation in therapy for cancerous diseases [28,35,36], a potential use of this therapeutic strategy in autoimmune and noncancerous inflammatory diseases remains unclear. This is especially intriguing as pharmacological co-inducers of Hsp70 expression (e.g., carvacrol) were able to downregulate the inflammation process in preclinical models of autoimmune arthritis [37,38,39]. Mechanistically, in contrast to Hsp90, the upregulation of intracellular Hsp70 inhibits the activity of NF-κB, a transcription factor that plays a key role in the inflammation process and autoimmunity [8]. In addition, DNA vaccines coding for Hsps including Hsp70 are considered as potential treatment of autoimmune disorders [40].

### Extracellular Hsp70 Activities

Bacterial and autologous Hsp70 may be released from normal and stressed cells and impact the host’s immune components belonging to the innate and acquired arms of the immune system. While data concerning the immunosuppressive activity of intracellular Hsp70 are generally consistent, with some exceptions, the role of extracellular Hsp70 in the inflammation process and the development of autoimmune diseases is still not fully determined, as scientists working in this field present often-conflicting data. These differences may be the result of interpretation errors that stem from presenting an incomplete picture of the immune responses to Hsp70, insufficient/inadequate quality/purity of protein preparations used in cell cultures, or simply due to fundamental disparity in the way in vitro and in vivo experiments are conducted. These apparently ambiguous activities may also be related to the ability of this chaperone to interact with multiple receptors that are displayed on different types of cells in the immune system. Finally, it is suggested that this dual nature corresponds to the mechanism of secretion/presentation. While Hsp70 presented by the antigen-presenting cells (APC) may activate regulatory T helper cells (Treg), thus promoting anti-inflammatory mechanisms, Hsp70 liberated from damaged (necrotic) cells can act as a damage-associated molecular pattern (DAMP) via Toll-like receptors 2 and 4 (TLR2 and TLR4), which stimulates proinflammatory reactions [5,8,9,10,41,42]. It should also be kept in mind that, despite the common features of autoimmune diseases, there are some substantial differences (involving different cells of the immune system) that may ultimately elicit the dual (pleiotropic) nature of extracellular Hsp70 and other Hsps. Multiple studies presenting the immunosuppressive activities of Hsp70-derived molecules are based on preclinical models of arthritis [43]. For instance, the research team of professor van Eden W. reported, in a very elegant way, that highly conserved Hsp70-peptide (HSP70-B29) used for active immunization of animals could be regarded as a potential treatment target for rheumatoid arthritis (RA) via induction of immunosuppressive Hsp70-specific Tregs [44]. The description of the dual role of extracellular Hsp70 in the development of autoimmune skin diseases is presented in later chapters and in Figure 2.

## 4. Hsp90 and Hsp70 as Potential Therapeutic Targets in Autoimmune Skin Diseases

The skin, which protects us from environmental and microbial insults thanks to physical and immunological barriers, is the largest organ of the human body. This protection is ensured by the skin’s unique anatomy and cellular composition, in particular, the network of immune cells including macrophages, dendritic cells, mast cells, γδ T cells, and innate lymphoid cells. During microbial skin infection, different activated T cell subpopulations, monocytes, and granulocytes may be additionally recruited to the skin to support host defense [45]. When the body overcomes an infection, the primary activity status of the immune cells is restored thanks to immunoregulatory mechanisms. Colloquially, autoimmunity may occur when the balance between the effector arm and the regulatory arm of the innate and adaptive immune systems is disturbed or improperly regulated. Therefore, the primary therapies for autoimmune skin diseases involve the use of immunosuppressive medications, including corticosteroids (applied both topically and systemically), methotrexate, or azathioprine, which inhibit the proinflammatory immune response. Currently, precisely targeted therapies relying on proinflammatory cytokine blockade (e.g., TNF, IL-17, and TSLP), cell depletion (e.g., B cells), blockade of intracellular signaling (e.g., JAK-STAT), or a costimulatory blockade (e.g., CD28/CD80-86/CTLA-4) are approved by the FDA or under clinical evaluation [46,47]. Despite the huge breakthrough made in the treatment of autoimmune skin diseases, currently available therapy still fails to bring a satisfactory effect in a number of cases. Hence, safer and more effective therapies that bring a sustained balance between pro- and anti-inflammatory responses are still desired.

There is growing evidence from preclinical studies confirming the contribution of either Hsp90 or Hsp70 to the development and therapy of autoimmune skin diseases, such as autoimmune bullous skin diseases, psoriasis, systemic lupus erythematosus, vitiligo, alopecia areata, or systemic sclerosis. Studies targeting Hsp90 or Hsp70 in autoimmune skin diseases based on preclinical and clinical studies are presented in Table 1.

### 4.1. Autoimmune Bullous Diseases

Autoimmune bullous diseases (AIBDs) belong to a relatively rare and potentially life-threatening organ-specific group of inflammatory skin diseases characterized by the presence of autoantibodies against various structural proteins of the skin present in desmosomes (e.g., pemphigus vulgaris-PV) and hemidesmosomes (e.g., bullous pemphigoid-BP and epidermolysis bullosa acquisita-EBA), or against epidermal/tissue transglutaminases present in Duhring disease (also known as dermatitis herpetiformis-DH). Despite understanding of the pathophysiology of AIBDs, in which both innate and adaptive mechanisms of the immune response are undoubtedly involved, treatment for this group of diseases remains a challenge, due to frequent relapses after the discontinuation of therapy, numerous side effects associated with using, e.g., corticosteroids, or due to the lack of a fully effective drug [67,68,69].

#### 4.1.1. Bullous Pemphigoid

Bullous pemphigoid (BP) is one of the most common types of AIBDs, characterized by the presence of tense blisters and tissue-bound autoantibodies directed to two hemidesmosomal structure proteins, namely, BP180 (specifically to its immunodominant region, NC16A) and BP230. In addition to the T-cell-dependent humoral autoreactivity, the importance of complement activation, neutrophils, macrophages, mast cells, and various proteases including neutrophil elastase and matrix metalloproteinases (MMPs) for blister formation are indicated [70].

It has been found that Hsp90 is accumulated/overexpressed in the perilesional skin of BP patients as compared to normal skin of healthy controls. Experimental approaches revealed that the accumulation of this chaperone in BP is mediated by the presence of circulating anti-BP180-NC16A IgG autoantibodies, since the levels of Hsp90 in circulation were significantly lower in affected patients compared to healthy controls and inversely correlated to the titer of anti-BP180-NC16A IgG in those patients. In addition, while Hsp90 expression was upregulated in activated (by BP serum) human keratinocytes (HaCaT), the presence of purified anti-BP180-NC16A IgG blocked the secretion of Hsp90 from HaCaT cells in vitro [71]. The role of the Hsp90 in BP has also been proven experimentally using an Hsp90 inhibitor—17-DMAG. The presence of this inhibitor in HaCaT cultures stimulated by anti-BP180-NC16A IgG (an in vitro approximation of BP ‘model’) led to the inhibition of NF-kB activation and a decreased expression/secretion of IL-8, which is one of the key chemokines in BP. In addition, 17-DMAG treatment was associated with Hsp70 induction in the cells [72].

#### 4.1.2. Epidermolysis Bullosa Acquisita

Epidermolysis bullosa acquisita (EBA) is an anti-type VII collagen (COL7) autoantibody-mediated autoimmune blistering skin disease with two major clinical subtypes, including mechanobullous or inflammatory variants, with the latter resembling BP [70]. In the BP-like experimental model of EBA, anti-COL7 IgG binding is followed by reactive oxygen species (ROS) generation and MMPs expression by neutrophils. Both directly lead to the degradation of the dermal–epidermal junction and blister formation [73].

Anti-Hsp90 therapy was examined in COL7-immunized mice using intraperitoneally applied 17-DMAG or nontoxic peptide TCBL-145. Both inhibitors ameliorated the clinical symptoms of EBA, suppressed anti-COL7 IgG, and reduced dermal neutrophilic infiltration. While total or antigen-specific plasma cells and germinal center B cells were unaffected by anti-Hsp90 treatment, human or mouse B cell as well as T cell activation were potentially suppressed by the inhibitors [48,49]. Mechanistically, 17-DMAG treatment led to the induction of HSF-1 and Hsp70 and reduced Th1 and Th17 frequencies in human activated B and T cell cultures, respectively [49,74]. In addition, anti-Hsp90 therapy led to the induction of regulatory B cells (Breg), the activation of which was proven ex vivo [49]. In a dose-dependent manner, 17-DMAG inhibited the detachment of the epidermis from the dermis in the human skin biopsy (cryosection assay) treated with anti-COL7 IgG-activated granulocytes, as well as inhibiting the production/release of ROS by activated human neutrophils. Finally, since extracellular Hsp90 has been found to interact with MMP2 and MMP12 in the sera of EBA patients, it may suggest that these basement-membrane-degrading enzymes are substrates of this chaperone and are thus potentially dependent on its activity [75]. The effectiveness of anti-Hsp90 therapy has also been demonstrated locally. Application of a less toxic geldanamycin analog, 17-AAG, directly to skin lesions attenuated clinical disease severity (both in prophylactic and therapeutic treatment) without skin or systemic toxicity in experimental EBA models. The therapy led to the reduction in skin infiltration by neutrophils and NF-κB activation, as well as decreases in MMP2, MMP9, MMP12, and Flii expression. In addition, topical 17-AAG application led to the induction of Hsp70 in the skin [50].

It has been frequently reported that Hsp70-derived peptides, used for active immunization of animals, could be regarded as a potential treatment target for RA via the induction of immunosuppressive mechanisms, which include IL-10 and Tregs [43]. Since blood levels of autologous inducible Hsp70 were found to be elevated in an experimental EBA mouse model [51], the role of this protein in EBA has yet to be determined. This disease is characterized by a different mechanism of development than RA. Surprisingly, Hsp70-treated EBA mice displayed more severe symptoms compared to untreated EBA mice. This effect was accompanied by increased levels of cutaneous MMP9 and circulating H_2_O_2_. ROS release assay using human granulocytes stimulated by EBA-specific immune complexes confirmed the proinflammatory properties of autologous Hsp70 [51]. Additionally, the humoral autoimmune response to Hsp70 has been observed to play a role in EBA. Circulating anti-Hsp70 IgG autoantibodies were significantly elevated in EBA patients as compared to healthy individuals and positively correlated with IFN-γ in patients. The importance of these newly discovered, pathologically relevant autoantibodies has been proven in vivo, since anti-Hsp70 IgG-treated EBA mice had a more intense clinical and histological disease activity [52]. The above outcomes suggest that both autologous Hsp70 and autoantibodies to Hsp70 display proinflammatory activities in the context of EBA development [51,52].

#### 4.1.3. Dermatitis Herpetiformis

Dermatitis herpetiformis (DH) is an autoimmune blistering skin manifestation of celiac disease that develops mostly in patients with latent gluten-sensitive enteropathy. It is manifested by the presence of gluten-induced IgA autoantibodies against epidermal (eTG) and tissue (tTG) transglutaminases [76]. In fact, circulating autoantibodies to Hsp60, Hsp70, and Hsp90 were found to be significantly elevated in DH patients (but not in BP and PV patients) in the active phase of the disease. Interestingly, remitted patients were characterized by a significant decrease in the level of anti-Hsps autoantibodies, as well as autoantibodies directed to eTG and tTG. These serological observations may suggest that autoantibodies to Hsps participate in the development and maintenance of DH, which suggests a potential novel use as disease biomarkers [77].

### 4.2. Psoriasis

Psoriasis is one of the most common chronic autoimmune skin conditions, characterized by scaly and itchy patches of reddened skin resulting from excessive proliferation and abnormal differentiation of epidermal keratinocytes caused by an impaired local and systemic immune responses, the presence of susceptibility alleles, and environmental factors [78]. There is a growing body of research suggesting that psoriasis is a systemic incurable disease where proinflammatory T helper cells, specifically the Th17 subpopulation, play an important effector role in initiating systemic inflammation [79]. Since proving that Hsp90 was involved in IL-17-mediated skin inflammation, the chaperone was found to be significantly upregulated in keratinocytes and mast cells in the lesional skin of patients with psoriasis [80]. Therefore, it has been described as a potential novel therapeutic target in this disease. In fact, the use of an Hsp90 inhibitor in the treatment of patients with psoriasis was discovered accidently during the first clinical trials of a new oral Hsp90 inhibitor (Debio 0932) in the treatment of oncological diseases, such as advanced solid tumors, lymphomas, and non-small cell lung cancer. Symptoms of psoriasis, which one of the oncological patients was suffering from, had gone into complete remission as a result of the experimental treatment administered during the trial. The confirmation of these unexpected results was provided in a psoriasis xenograft transplantation model. In this mouse model, oral administration of Debio 0932 to animals significantly arrested both the development of the clinical manifestation of psoriasis and histological parameters, such as epidermal thickness, as well as psoriasis pattern and vessel scores [53]. In vitro studies using another Hsp90 inhibitor (RGRN-305) on psoriasis-like inflammatory response in human keratinocytes further confirmed the contribution of this chaperone to the development of psoriasis. RGRN-305 significantly reduced the IL-17A- and TNFα-induced expression of CCL20, NFKBIZ, IL-36G, and IL-23A in human keratinocyte cultures [81]. These preclinical observations have resulted in the use of this experimental therapy in clinical trials. A phase Ib proof-of-concept study has been recently launched to evaluate the safety and efficacy of RGRN-305 in the treatment of plaque psoriasis [57]. It has been reported that six of the eleven psoriasis patients enrolled in this clinical study responded positively to RGRN-305 treatment, as a 71–94% reduction in the Psoriasis Area and Severity Index (PASI) in RGRN-305-treated patients was noted. While five out of eleven patients were considered non-responders, no serious side effects were reported. Treatment with RGRN-305 resulted in a marked inhibition of the IL-23, TNF-α, and IL-17A signaling pathways and the normalization of both histological changes and the gene expression profiles of psoriasis, further supporting that Hsp90 may serve as a novel target in the treatment of psoriasis [57].

Hsp70 has also been considered as a therapeutic target in psoriasis. This claim was tested using a well-described imiquimod (IMQ)-induced mouse model of psoriasis-like skin inflammation. Interestingly, topical application of either Hsp70 inhibitors or plant-derived Hsp70 protein contributed to a significant inhibition of clinical and histological symptoms and a modulation of the selected disease-associated cytokines, including IL-17A, IL-4, IL-5, TNF-α, IL-22, and IL-23 [54,55]. These distinct therapeutic approaches (i.e., inhibition of Hsp70 activity or topical Hsp70 application) had surprisingly convergent clinical outcomes. It must therefore be clarified whether the chaperone activity of intracellular Hsp70 or the immunogenicity of the extracellular fraction of this protein is crucial in the development of psoriasis. Given the assumption that immunization with Hsp70 expanded Treg and ameliorated autoimmune arthritis in a model [82], the role of Hsp70-based immunization therapy on psoriasis-like skin inflammation (IMQ-induced mouse) was evaluated. Immunization of naïve BALB/c mice (two weeks prior to psoriasis induction) with autologous or plant Hsp70 resulted in decreased PASI and histological severity. Mechanistically, therapy with plant-derived Hsp70 led to the expansion of two populations of Tregs, i.e., CD4^+^FoxP3^+^ and CD4^+^CD25^+^. A functional assay revealed that concomitantly induced circulating anti-Hsp70 IgG in the immunized animals could also inhibit disease progress, since passive transfer of anti-Hsp70 IgG led to the attenuation of disease activity and the inhibition of Th17 frequencies in the spleens. Antiproliferative/immunosuppressive effects of Hsp70 on keratinocytes/T cells were also directly confirmed in cell culture experiments [56].

### 4.3. Vitiligo

Vitiligo is an autoimmune T-cell-dependent depigmenting disorder resulting from the loss of melanocytes in the epidermis [83]. It has been initially reported that 4-tertiary butyl phenol (4-TBP)-treated vitiligo PIG3V melanocytes may mediate the activation of disease-effector dendritic cells (DCs) through the release of Hsp70 [84]. Further, it has been shown that using human- and mouse-derived inducible Hsp70 in vaccination (plasmids encoding Hsp70s) accelerated depigmentation in a mouse model of autoimmune vitiligo [58], representing a potential therapeutic target [85]. Moreover, two independent studies have found cutaneous upregulation of Hsp70, which was associated with the development of vitiligo in patients [86,87]. Additionally, using an immortalized human vitiligo melanocyte cell line, both cytoplasmic Hsp70 and Hsp90 were found to be translocated into apoptotic bodies along with autoantigens (e.g., tyrosinase-related protein 1 or cleavage nuclear membrane antigen Lamin A/C) associated with vitiligo [88].

### 4.4. Alopecia Areata

Alopecia areata (AA) is a chronic inflammatory CD8^+^ T cells-mediated disease characterized by an autoimmune reaction to hair follicles, which consequently leads to the loss of hair in focal regions, the complete scalp including eyelashes and eyebrows, or even the entire body [89]. Quantitative proteomic analysis revealed that Hsp90 and Hsp70 chaperones were among 104 downregulated proteins found in lesional compared to non-lesional skin biopsies of AA patients [90]. Further studies demonstrated that Hsp90 and Lamin A/C interact with each other, and both play an essential role in the growth, migration, and self-aggregation of dermal papilla cells and can be linked to AA. It is suggested that the disruption of such interactions may contribute to the pathogenesis of AA via the dysfunction of dermal papilla cells [91].

Preclinical studies revealed not only an enhanced skin Hsp70 expression in C3H/HeJ mice strain that developed AA spontaneously [92], but also that the blockade of Hsp70 by quercetin proved an effective treatment for AA in this model [59].

### 4.5. Systemic Lupus Erythematosus

Systemic lupus erythematosus (SLE) is a prototype autoimmune disease characterized by antinuclear antibodies, immune complex deposition, and heterogeneous clinical manifestations involving various organs and tissues, e.g., skin, joints, or kidneys. The disease manifests with various abnormalities in the phenotypes and functions of the innate and acquired immune cells [93]. The role of Hsp90 in SLE and its clinical relevance has been confirmed independently by several research teams. Higher expression of Hsp90 as well as enhanced circulating levels of soluble Hsp90 and anti-Hsp90 autoantibodies were associated with disease progression. Clinical improvement and multimodal regulatory effects of Hsp90 inhibition on, e.g., cell signaling, proinflammatory cytokine secretion, dsDNA antibodies, proteinuria, or lymphocytes activity (e.g., expansion of Treg or reduction in pathogenic T and B cell lineage populations) in SLE was revealed using experimental animal models. Taken together, Hsp90 contributes to inflammation and SLE progression, and therefore, targeting of its expression/activity may be a viable treatment for SLE [60,61,62,63,94,95,96,97,98,99,100].

Independent approaches revealed Hsp70 gene polymorphisms associated with SLE pathogenesis [101,102]. In addition, the expression of Hsp70 (HSPA1A) was significantly upregulated in patients with SLE and positively linked with the disease-specific autoantibodies in patients [103]. Preclinical studies, however, indicate a dual role for Hsp70 in the development of SLE. On the one hand, vaccination with DNA encoding Hsp70 suppressed a murine model of SLE via the induction of tolerogenic immune responses and marked suppression of anti-dsDNA antibody production, reduction in renal disease, and anti-inflammatory responses [64]. On the other hand, blockade of HSC70/Hsp73 chaperone expression via the P140 peptide displayed protective properties in MRL/lpr lupus-prone mice by decreasing autoreactive T cell priming and signaling [104].

### 4.6. Systemic Sclerosis

Systemic sclerosis (SSc) is a rare systemic autoimmune disease, one of the connective tissue diseases, characterized by widespread skin (scleroderma) and internal organ fibrosis, disturbances of innate and acquired immune responses, as well as vascular abnormalities [105]. Multiple observations provide evidence for the contribution of Hsp90 in the development of skin fibrosis [106]. It has been found, for instance, that increased expression of Hsp90 in the skin of patients with SSc is critical for TGF-β signaling and that a pharmacological blockade of Hsp90 inhibited the profibrotic effects of TGF-β in cultured fibroblasts and in animal models of SSc [65]. Importantly, the chaperone’s extracellular presence, i.e., elevated serum Hsp90 levels, were associated with increased systemic inflammation, worse lung functions, and skin involvement in SSc patients [107]. Likewise, increased levels of circulating Hsp70 were associated with pulmonary fibrosis, skin sclerosis, renal vascular damage, oxidative stress, and inflammation in SSc patients, suggesting that extracellular Hsp70 may be a useful serological marker for evaluating both cellular stresses and disease severity in SSc patients [108].

### 4.7. Atopic Dermatitis

Atopic dermatitis (AD) is one of the most common chronic inflammatory skin diseases (prevalence: 15–30% in children and 2–10% in adults), characterized by intense itching and recurrent skin lesions. AD is a chronic and incurable immune-mediated disease that can be controlled by the use of topical emollients, calcineurin inhibitors, or corticosteroids [109]. Since immune responses to self-proteins have been observed in AD patients, it cannot be excluded that an autoimmune response plays an important role in the progression of this disease [110]. Even though AD is not currently classified as an autoimmune disease, recent observational studies revealed the potential association between AD and autoimmune disorders. Systematic review and meta-analysis studies showed a significant association of AD with various autoimmune diseases, including AA, celiac disease, Crohn’s disease, RA, SLE, ulcerative colitis, and vitiligo [111].

Highly immunogenic extracellular Hsp90 can activate the humoral immune response driving the generation of circulating anti-Hsp90 autoantibodies that were found to be elevated in several autoimmune disorders [8,9]. In fact, one study found circulating Hsp90 to be significantly elevated in AD patients compared to healthy controls and positively correlated with the severity of AD (SCORAD; Scoring Atopic Dermatitis). In the same study, anti-Hsp90 IgE serum positivity was characterized for about 50% of AD patients and less than 3% of healthy controls. These results suggest a possible role of the extracellular Hsp90 and anti-Hsp90 IgE autoantibodies in the development of AD, as well as providing potential novel disease biomarkers [112].

It has been found that the upregulation of Hsp65 and Hsp72/73 in skin lesions of patients with AD was positively associated with the disease’s severity [113]. Another study revealed that elevated levels of anti-Hsp70 antibodies were associated with metal allergy in AD patients [114]. On the other hand, subcutaneous administration of recombinant Hsp70 to mice with an OVA-induced AD-like phenotype ameliorated disease severity and cellular skin inflammation via the induction of systemic Th1- and inhibition of Th2-like immune responses, as well as the inhibition of cutaneous TSLP expression [66].

## 5. Prospective

Extensive preclinical studies using Hsp-based vaccines or the inhibition of Hsp expression/activity in experimental autoimmune animal models have been validated in the first clinical trials and proof-of-concept studies, giving hope for the development of next-generation drugs. Promising clinical trials have been performed in patients with rheumatoid arthritis using Hsp40-derived dnaJP1 (a highly conserved 15 aa peptide), as well as full-length Hsps, such as immuno-globulin binding protein (BiP, belonging to the Hsp70 family) and Hsp10. In parallel, the treatment of newly diagnosed type 1 diabetes patients with Hsp60-derived DiaPep277 peptide successfully entered phase III clinical trials [8,9,43,115,116]. In addition, the latest clinical trials confirm the safety and efficacy of Hsp90 inhibition by RGRN-305 in the treatment of plaque psoriasis [57]. Taken together, both therapeutic approaches, i.e., modulation of Hsp activity/expression and Hsp-based vaccines, seem to represent a new promising direction in the treatment of autoimmune skin diseases. To fully validate this approach, long-term efficacy and safety, optimal dosage, and the use of Hsp-derived proteins/chaperone inhibitors in combination with current therapy all require further research. Therefore, further and more extensive studies using animal experimental models should be performed to determine the roles of such chaperones in the pathogenesis of autoimmune skin diseases. It is also worth noting that studies suggesting the involvement of Hsp90/70 in autoimmune/inflammatory diseases based solely on observations of changes in the level of protein expression in inflamed tissues may not be sufficient to draw a final conclusion. Upregulation of Hsp90/70 and the correlations between the levels of these chaperones both inside and outside the cell with selected clinical parameters cannot unequivocally determine their contribution in the pathological process since the presence of such associations may result from compensation mechanisms. Therefore, functional assays or animal experimental models should be implemented to determine the role of such chaperones in the pathogenesis of autoimmune diseases. This is particularly important in the study of Hsp90/70, whose rapid and efficient expression in a cell exposed to multiple stressors is a key element of first response, enabling the repair of any damage that has occurred.

## Figures and Tables

**Figure 1 biomolecules-12-01153-f001:**
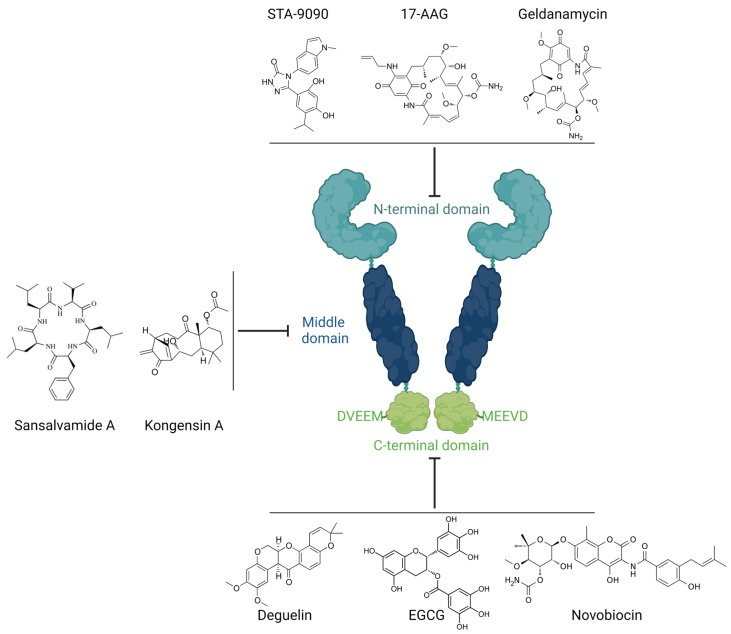
Schematic representation of the structural domains of Hsp90 along with their specific inhibitors, categorized by binding sites. Created with BioRender.com (accessed on 9 August 2022).

**Figure 2 biomolecules-12-01153-f002:**
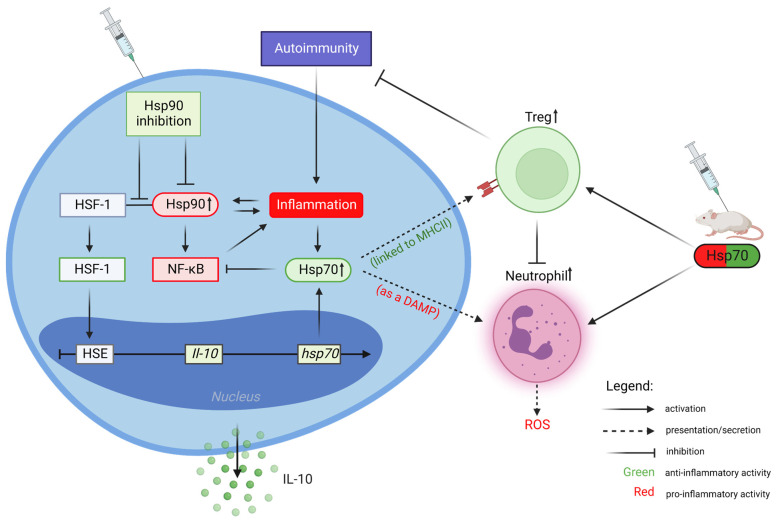
Contribution of Hsp90 and Hsp70 to the immune response and their significance in the therapy of autoimmune diseases. Inflammation leads to Hsp90 upregulation and, vice versa, Hsp90 promotes proinflammatory signaling. Blockade of Hsp90 activates heat shock factor 1 (HSF-1), which upregulates the expression of Hsp70 and IL-10. Stress stimuli or Hsp90 inhibition lead to the induction of Hsp70, which downregulates NF-κB activation. In parallel, Hsp70 presented by major histocompatibility complex (MHC) class II molecules activates disease-modulating (immunosuppressive) T regulatory cells (Treg). On the other hand, acting as a damage-associated molecular pattern (DAMP), extracellular Hsp70 activates neutrophils and promotes the secretion of reactive oxygen species (ROS). Hsp70-based immunization stimulates either pro- or anti-inflammatory immune responses/cells. Created with BioRender.com (accessed on 11 August 2022).

**Table 1 biomolecules-12-01153-t001:** Overview of studies targeting Hsp90 or Hsp70 in autoimmune skin diseases based on preclinical and clinical observations.

Disease	Animal Model/ Clinical Observation	Target	Inhibitor	Outcome	Literature
**Epidermolysis bullosa** **acquisita**	COL7 or anti-COL7 IgG immunized mouse models	Hsp90	TCBL-145	Clinical and histological improvement	[48]
			17-DMAG		[48,49]
			17-AAG		[50]
		Hsp70	None	Hsp70- or anti-Hsp70 IgG-treated EBA mice had more intense disease activity	[51,52]
**Psoriasis**	Mouse xenograft transplantation model	Hsp90	Debio 0932	Clinical and histological improvement	[53]
	Imiquimod-induced mouse model	Hsp70	Myricetin Quercetin Ellagic acid	Clinical and histological improvement	[54]
		Hsp70	None	Topically applied Hsp70 ameliorated disease activity	[55]
		Hsp70	None	Hsp70-based immunization ameliorated disease activity	[56]
	Phase I–II evaluation of safety and efficacy	Hsp90	Debio 0932	Clinical improvement during unrelated (oncological diseases) clinical trial	[53]
	Phase Ib proof-of-concept study	Hsp90	RGRN-305	Clinical and serological improvement	[57]
**Vitiligo**	Mouse model of autoimmune vitiligo	Hsp70	None	Human- and mouse-derived inducible Hsp70-vaccinated mice displayedaccelerated depigmentation	[58]
**Alopecia** **areata**	C3H/HeJ spontaneous mouse model of AA	Hsp70	Quercetin	Clinical and histological improvement	[59]
**Systemic** **lupus** **erythematosus**	MRL/lpr mouse model of SLE	Hsp90	17-AAG	Clinical and functional improvement	[60]
			STA-9090		[61]
			17-DMAG		[62]
	(NZB × NZW)F1 mouse model of SLE	Hsp90	None	Vaccination with DNA encoding Hsp90 protected from murine lupus	[63]
		Hsp70	None	Vaccination with DNA encoding Hsp70 led to disease suppression	[64]
**Systemic** **sclerosis**	Mouse model of Ssc	Hsp90	17-DMAG	Histological and functional improvement	[65]
**Atopic** **dermatitis**	OVA-induced mouse model	Hsp70	None	Subcutaneous administration of recombinant Hsp70 led to clinical, histological, and serological improvement	[66]

## Data Availability

Not applicable.

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
