# Peer review of "Heat Shock Protein 90 (Hsp90) and Hsp70 as Potential Therapeutic Targets in Autoimmune Skin Diseases"

_biomolecules, 2022, doi:10.3390/biom12081153_

Round 1
Reviewer 1 Report
The authors provide a detailed, informative and well written review about Hsp90 and Hsp70 as potential therapeutic targets in autoimmune skin diseases.
Major comments
1) Figure 1: Is extracellular Hsp90 missing? Or does the Hsp70 in the figure outside of the cell only represents the immunization of mice? The figure suggests that inhibition of Hsp90 results in HSF-1 not being released, please improve.
2) Page 3, last paragraph: I suggest to write a conclusion in the end of the manuscript and to add the information from the last paragraph on page 3 there
3) Page 4, line 174: the name of the protein is BP180 and NC16A the main immunodominant region, please change
4) Page 5, line 184…: did BP serum also blocked the Hsp90 secretion? Can you provide more details?
5) Page 7, line 293: please provide more information
6) Page 7, line 296: which mice? Please provide more details.
7) Page 7, line 310: what is used for vaccination?
8) Page 7, line 314: … be translocated into apoptotic bodies… in keratinocytes in culture or in the mouse model? Please, provide more information.
9) Page 7, line 315: can you provide examples for the autoantigens?
10) Page 7, 4.4 Alopecia areata: please, include a sentence about the clinical picture of the patients. Please provide more information about the link of Hsp90 to AA
11) Page 9, Summary: The Summary seems to be more “prospective” or “conclusion”. I would suggest to change the heading “summary” to a more appropriate title or insert a more appropriate summary
Minor comments
1) Page 3, line 91: HSP70 is…(delete the)
2) Page 3, line 94: “… contrast to Hsp90, upregulation of the intracellular Hsp70 inhibits…(delete the or write Hsp 70 familiy); same in line 102
3) Page 4, line 146: I think it would make sense to introduce some brackets “ .. medications including corticosteroids (applied both applied topically and systemically), methotrexate, or azathioprine, that …
4) Page 5, line 200: please explain the abbreviation TCBL-145
5) Page 5, line 222: …in an experimental..
6) Page 5, line 230: … confirmed the pro-inflammatory properties …
7) Page 7, line 293: … which implies that further clarification …
8) Page 7, line 307: please explain the abbreviation 4-TBP
9) Page 7, line 332: The role
Reviewer 2 Report
This review has a very good approach to tap the role of HSP 70 & 90 in autoimmune skin diseases.
After reviewing this article, I am reaching conclusion, this review article has lots of scope of improvement.
The current review is based on HSP 70 & 90, a potential therapeutic target for autoimmune diseases, but the description of inhibitors has not been discussed very well in this review.
Authors should have enriched this article with a complete list of current, probable and experimental chemical and natural inhibitors, along with chemical structures, mode of action and details of ongoing clinical trails on the inhibitors.
Even authors can also discuss a special point of non-coding RNA-based inhibitors like siRNA and miRNA etc, that have the capacity to modulate the function of HSP 70 & 90.
The structural elucidation and difference between both HSPs should be discussed, for the better selection of inhibitors.
I appreciate the authors for good representation of figure 1, but the elucidation part of this review should be improved.
Instead of providing the big paragraphs, authors can convert them into a pictorial or tabular format, for better understanding and readability.
Reviewer 3 Report
This review is very comprehensive and provides a great overview on the importance of Hsps in autoimmune skin diseases. The authors have listed the clinical trials throughout the paper which clearly highlights the importance of these proteins but if a table summarizing these trials and the publications could be made it would be a very useful resource for the readers.
Round 2
Reviewer 1 Report
Thank you very much for your point-by point reply. I have no further comments.
Reviewer 2 Report
I would recommend to accept this review article for the publication.